

# 3D-GloBFP: the first global three-dimensional building footprint dataset

Yangzi Che[1], Xuecao Li[2], Xiaoping Liu[1,3], Yuhao Wang[1], Weilin Liao[1], Xianwei Zheng[4], Xucai Zhang[5], Xiaocong Xu[1], Qian Shi[1], Jiajun Zhu[1], Hua Yuan[3,6], Yongjiu Dai[3,6]

[1]Guangdong Key Laboratory for Urbanization and Geo-simulation, School of Geography and Planning, Sun Yat-sen University, Guangzhou, 510275, China
[2]College of Land Science and Technology, China Agricultural University, Beijing, 100083, China
[3]Southern Marine Science and Engineering Guangdong Laboratory (Zhuhai), Zhuhai, 519082, China
[4]The State Key Lab. LIESMARS, Wuhan University, Wuhan, 430079, China
[5]Department of Geography, Ghent University, Ghent, 9000, Belgium
[6]School of Atmospheric Sciences, Sun Yat-sen University, Guangzhou, 510275, China

*Correspondence to*: Xiaoping Liu (liuxp3@mail.sysu.edu.cn)

**Abstract.** Understanding urban vertical structures, particularly building heights, is essential for examining the intricate interaction between humans and their environment. Such datasets are indispensable for a variety of applications, including climate modeling, energy consumption analysis, and socioeconomic activities. Despite the importance of this information, previous studies have primarily focused on estimating building heights regionally on a grid scale, often resulting in datasets with limited coverage or spatial resolution. This limitation hampers comprehensive global analyses and the ability to generate actionable insights on finer scales. In this study, we developed a global building height map (3D-GloBFP) at a building footprint scale by leveraging Earth Observation (EO) datasets and advanced machine learning techniques. Our approach integrated multisource remote sensing features and building morphology features to develop height estimation models using the eXtreme Gradient Boosting (XGBoost) regression method across diverse global regions. This methodology allowed us to estimate the heights of individual buildings worldwide, culminating in the creation of the first global three-dimensional (3-D) building footprints (3D-GloBFP). Our evaluation results show that the height estimation models perform exceptionally well on a worldwide scale, with $R^2$ ranging from 0.66 to 0.96 and root mean square errors (RMSEs) ranging from 1.9 m to 14.6 m across 33 subregions. Comparisons with other datasets demonstrate that our 3D-GloBFP closely matches the distribution and spatial pattern of reference heights. Our derived 3-D global building footprint map shows a distinct spatial pattern of building heights across regions, countries, and cities, with building heights gradually decreasing from the city center to the surrounding rural areas. Furthermore, our findings indicate the disparities in built-up infrastructure (i.e., building volume) across different countries and cities. China is the country with the most intensive total built-up infrastructure ($5.28\times10^{11}$ m³, accounting for 23.9 % of the global total), followed by the United States ($3.90\times10^{11}$ m³, accounting for 17.6 % of the global total). Shanghai has the largest volume of built-up infrastructure ($2.1\times10^{10}$ m³) of all representative cities. The derived building-footprint scale height map (3D-GloBFP) reveals the significant heterogeneity of urban built-up environments, providing valuable insights for



studies in urban socioeconomic dynamics and climatology. The 3D-GloBFP dataset is available at https://doi.org/10.5281/zenodo.11319913 (Building height of the Americas, Africa, and Oceania in 3D-GloBFP) (Che et al., 2024a), https://doi.org/10.5281/zenodo.11397015 (Building height of Asia in 3D-GloBFP) (Che et al., 2024b), and https://doi.org/10.5281/zenodo.11391077 (Building height of Europe in 3D-GloBFP) (Che et al., 2024c).

## 1 Introduction

Quantifying the three-dimensional (3-D) building structure is essential in understanding human-natural ecosystems and achieving sustainability goals. The World Cities Report 2022 reveals that urban areas already accommodate 55 % of the global population, and the figure is expected to grow to 68 % by 2050 (United Nations Human Settlements Programme, 2022). Under the background of advancing global urbanization, burgeoning populations pose challenges and opportunities to land-use efficiency, making vertical urban growth a critical land-use pattern (Chen et al., 2024; Chen et al., 2020). Various urban

functions have also given rise to distinct 3-D spatial forms within cities (Demuzere et al., 2022). Specifically, commercial central areas show a dense concentration of high-rise buildings, residential zones are characterized by rows of relatively tall buildings and urban villages are distinguished by dense clusters of low-rise structures (Chen et al., 2023b). In this context, the accurate three-dimensional mapping of urban areas is a crucial objective for achieving sustainable and resilient cities. Building height, as the vertical structure of buildings, can depict the urban vertical morphology, which reflects the biophysical and

social-economical properties of the cities and supports a variety of urban studies, including climate mitigation, carbon emission, living conditions, socioeconomic modeling (Pappaccogli et al., 2020; Xu et al., 2021; Shao et al., 2023; Shang et al., 2020), and so on. For instance, accurate measurement of building heights is essential for determining the urban underlying surface, serving as critical urban parameters in urban climate models to simulate and understand the climate conditions within urban areas (Sun et al., 2021). Simultaneously, 3-D building datasets help assess built-up infrastructure spaces and further contribute

to the 2023 agenda for Sustainable Development Goals (SDGs) aimed at providing adequate, safe, and affordable houses for all (Liu et al., 2024). Moreover, building heights provide demographic insights and help delineate functional zones within cities, thereby enhancing the estimation of energy use and carbon emissions (Ding et al., 2022).

While Earth Observations have been generally used in 3-D building mapping, the estimation of building height is still limited either in spatial resolution or coverage. High-resolution optical images, Synthetic Aperture Radar (SAR), and Airborne Light

Detection and Ranging (LiDAR) products are the commonly used datasets for extracting building height information in the urban domain. High-resolution optical satellite images can provide texture and shadow details within urban areas, which can be applied to building height estimation (Cao and Huang, 2021; Liasis and Stavrou, 2016; Chen et al., 2023a). However, the accuracy is limited by the quality of images, and the effectiveness is reduced in densely built areas (e.g., Central Business District (CBD)) where building shadows are overlaid with other objects (Cai et al., 2023). Alternatively, SAR images can

reflect the scattering mechanism of buildings through the backscatter coefficients, which are related to building structure (Koppel et al., 2017). A variety of studies have been carried out using SAR data for built-up height estimation. Li et al. (2020b)





and Zhou et al. (2022) developed an approach to estimate building height using the dual-polarization information (i.e., VV and VH) from the Sentinel-1 dataset, while the reliability of height estimation under fine-scale (i.e., less than 500m) is constrained due to the "bounce scattering" effect (Li et al., 2020b). Instead, LiDAR is regarded as the most reliable data source

for obtaining building height because it can directly capture the rooftop coordinates from the returned signal (Li et al., 2020a; Park and Guldmann, 2019). However, the LiDAR dataset is scarce and scattered, making it difficult to apply over larger areas (Ma et al., 2023).

Although multisource datasets offer broader coverage of building height estimation, globally fine-scale (i.e., building scale) building height datasets are still absent, disregarding the spatially explicit heterogeneous building form. Current researchers

proposed methods based on Digital Surface Models (DSM) and statistical modeling to estimate building heights, enhancing the coverage of height estimation. Firstly, widely available digital elevation models (i.e., ALOS DSM and TanDEM-X) provide information for height estimation. Esch et al. (2022) acquired global building heights at 90 m resolution by computing the difference between local maximum and minimum within built-up areas using the SAR-derived TanDEM-X. However, uncertainties may arise in rugged regions (Huang et al., 2022). Additionally, Huang et al. (2022) used slope correction to

mitigate slope effects and derived building height in China. However, the 30 m dataset is also affected by a mixed object problem (i.e., one pixel contains both building and surrounding terrain), which smooths the height edge and consequently increases the inaccuracy of building height estimations (Esch et al., 2022). Secondly, the statistical modeling method can obtain continuous building height estimation at the regional (i.e., national or urban agglomeration) scale by training machine learning models with multiple explanatory features. Frantz et al. (2021) and Wu et al. (2023) integrated Sentinel-1 and Sentinel-2

datasets and extracted building height based on the machine learning method, confirming the effectiveness of fusing SAR and optical datasets. Arehart et al. (2021) combined various physical morphological features of buildings (e.g., area, compactness, and radius) to derive building heights in the United States, providing evidence of the correlation between morphological features and height. Li et al. (2022) generated a global-scale building height map at 1 km resolution by utilizing optical, SAR, and auxiliary geospatial data (e.g., GDP and road networks) based on a random forest model. Moreover, Ma et al. (2023) fused

height metrics from Global Ecosystem Dynamics Investigation (GEDI) and other explanatory features to obtain the building height in the Yangtze River Delta region at 150 m resolution. Nevertheless, due to the complexity of urban functions and diverse landscapes, spatially neighboured buildings may vary significantly in height. As a result, the grid resolution height data (e.g., 1 km) may be insufficient to accurately describe the 3-D spatial structure of buildings, leading to a loss of spatial information (Li et al., 2024b). Besides, raster datasets tend to blur building boundaries when representing the building shapes,

lacking the precision of vector footprints in representing the 3-D morphologies of buildings. Notably, there is currently no global dataset that reflects the height of building footprints.

To fill these gaps, we developed the first global dataset at individual building scale (3D-GloBFP) using open-access multisource datasets based on machine learning methods. The 3D-GloBFP datasets delineate the 3-D morphology of each building worldwide, capturing the 3-D spatial patterns of buildings in cities of various scales across the world. Specific

objectives of this study include: (1) integrate and preprocess the multisource remote-sensing datasets and morphology features





of building vectors; (2) develop the height estimation model in different subregions; (3) produce a building-scale height map globally in 2020, and (4) analyze the built-up infrastructures in global countries and cities. The remainder of this paper describes the adopted datasets (Sect. 2), the estimation method (Sect. 3), the results and discussion (Sect. 4), the data availability (Sect. 5), and conclusions (Sect. 6).

## 2 Datasets

### 2.1 Building footprint datasets

We derived the global building footprints using datasets from Microsoft building footprints (Microsoft, 2018) and building boundaries in Shi et al. (2024). The Microsoft building dataset provided 1.3 billion building footprints in the world around the year 2020. This dataset was derived from high-resolution satellite imageries using Deep Neural Networks (DNNs) and polygonization approaches. The derived building footprints in the Microsoft dataset are highly consistent with the boundary of individual buildings, with average precision and Intersection over Union (IoU) around 95 % and 65 %, respectively. Given that some regions in East Asia (e.g., China, North Korea, and South Korea) were not included in Microsoft building footprints, we used building footprints generated by Shi et al. (2024) as an alternative. Shi et al. (2024) extracted these building footprints based on high-resolution imageries using deep learning approaches with stable accuracy in different cities (i.e., the precision and recall in cities exceed 80 %). These two open-source datasets provided a complete building boundary dataset covering the globe, with good quality to support our research.

### 2.2 Building height datasets

We collected building footprint data with height information from ONEGEO Map (https://onegeo.co/data/), Microsoft building footprints (Microsoft, 2018), Baidu Maps (https://map.baidu.com/), and EMU Analytics (https://www.emu-analytics.com/) to ensure maximum coverage of reference building height across all regions globally (Fig. 1). ONEGEO Map integrates data from over 40 sources, including OpenStreetMap, USGS, and Google Open Buildings, offering comprehensive building height records for various regions worldwide. To obtain a more thorough and densely covered reference building height dataset, we supplemented it with the Microsoft dataset in the United States and the Baidu dataset in China. Microsoft building height released in 2018 provides the height of buildings in 44 states, where only a small fraction is located in the city center containing height attributes (i.e., only 2 % of buildings have height records in New York States). In addition, Baidu Map height datasets provide the height information in individual vector form in the core built-up areas in cities. This height dataset widely covers cities in China (i.e., metropolitans, all the capital provinces, big cities, and some small cities) (Fig. S1), which helps to ensure the robustness of the model in predicting building heights across the country. For example, the Baidu dataset includes 603,007, 443,436, and 23,980 individual buildings of different heights in Beijing, Foshan, and Ganzhou, respectively. The height dataset from the Baidu service is consistent with the actual building height, with an accuracy of 86.78 % and a mean deviation of approximately 1.02 m, as reported by Liu et al. (2021). We also used the building height dataset from EMU Analytics in



England. The EMU Analytics height dataset includes nearly 12 million building footprints, with building height calculated from 1 m resolution LiDAR images.

Our combined reference height dataset covers most regions worldwide, providing a comparatively reliable training and testing dataset for estimating building heights in various cities and regions globally (Fig. 1). Overall, reference data for building heights is abundant in major global cities. The reference building height data in the United States and China is plentiful in quantity and widely distributed. In Europe, reference building data covers various areas, from city centers to rural areas. Reference building height data is also relatively abundant in Central America, South America, and Western Asia. Building height data for Southeast Asia, North Asia, Africa, and Australia can basically cover all regions.

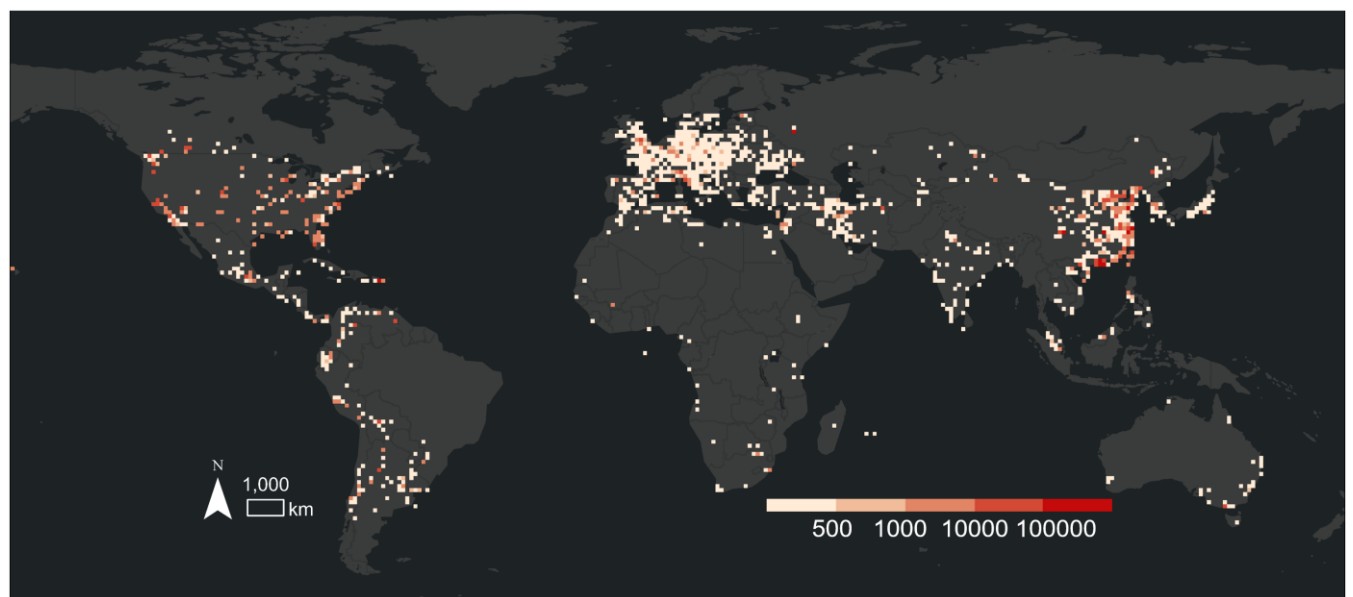


**Figure 1. Building with height properties at 0.1°×0.1° scale.**

## 2.3 Multisource remote sensing datasets

We integrated SAR images, optical images, terrain images, and images reflecting population and socioeconomic activities to estimate building height benefit from a wealth and easily accessible imageries provided by Google Earth Engine (Table 1). To

obtain the heights of buildings in 2020, we primarily used the multisource datasets from 2020, supplemented by imagery from adjacent years, to achieve seamless global coverage. Sentinel-1 mission consists of two polar-orbiting satellites performing C-band synthetic aperture radar imaging, allowing them to acquire images in all weather conditions. We collected the Ground Range Detection (GRD) type high-resolution (10m) images with dual polarization (i.e., VV, VH) in Sentinel-1 datasets, as the backscatter coefficient in GRD images are sensitive to surface roughness and can reflect the buildings' structure. We also used

variables from optical images (i.e., Sentinel-2) as the input of our height estimation model. The Copernicus Sentinel-2 mission includes a constellation of two polar-orbiting satellites, supporting the monitoring of the Earth's surface conditions. We used



Band2 (Blue), Band3 (Red), Band4 (Green), and Band8 (Near-infrared) in Sentinel-2 in our model at 10 m resolution. The radiation of visible bands is correlated with the extent of impervious surfaces and the internal environment within urban domains (Yuan and Bauer, 2007). The near-infrared band can effectively provide information on building heights by reflecting

the thermal radiation capability of the surface material. Furthermore, we collected terrain datasets (i.e., the Digital Elevation Model (DEM) from the Shuttle Radar Topography Mission (SRTM) at 30m resolution and the Digital Surface Model (DSM) at 30m from the Advanced Land Observing Satellite (ALOS)) to represent the physical properties of urban domains. DSM data provides vertical information about surface objects, which is helpful for extracting building heights. Primarily, the difference between DSM and DEM (nDSM) directly reflects the vertical height of surface objects. In addition, we used other

datasets to expand the number of features and provide auxiliary information on building height, including the Phased Array type L-band Synthetic Aperture Radar (PALSAR), WorldPop, and Visible Infrared Imaging Radiometer Suite (VIIRS) Day/Night Dataset. Population and nighttime light data are proven to be related to the building structure.

**Table 1. Multiple sources of datasets used in our study**

| Category | Datasets | Resolution | Acquisition time | Provider | Link | Reference |
|---|---|---|---|---|---|---|
| SAR | Sentinel-1 (VV, VH) | 10 m | 2019-2021 | European Union/ESA/ Copernicus | https://earth.esa.int/ | Koppel et al. (2017); Li et al. (2020b) |
|  | PALSAR (HH, HV) | 25 m | 2020 | JAXA EORC | https://www.eorc.jaxa.jp/ALOS/ | Wu et al. (2023) |
| Optical | Sentinel-2 (band2, band3, band4, band8) | 10 m | 2020 | European Union/ESA/ Copernicus | https://earth.esa.int/ | Frantz et al. (2021); Lyu et al. (2024) |
| Terrain | DEM | 30 m | 2000 | NASA/USGS/JPL-Caltech | https://cmr.earthdata.nasa.gov/ | Huang et al. (2022); Geiß et al. (2019) |
|  | DSM | 30m | 2006-2011 | JAXA Earth Observation Research Center | https://www.eorc.jaxa.jp/ALOS/ |  |





| | | | | | | |
|---|---|---|---|---|---|---|
| | CDEM | 0.75 arc-seconds | 1945-2011 | Natural Resources Canada | https://open.canada.ca | / |
| | WorldPop | 92.77 m | WorldPop | WorldPop | https://www.worldpop.org/ | Li et al. (2020a) |
| Social-economical | Nighttime light | 463.83 m | Earth Observation Group, Payne Institute for Public Policy, Colorado School of Mines | Earth Observation Group, Payne Institute for Public Policy, Colorado School of Mines | https://payneinstitute.mines.edu/ | Yu et al. (2022); Wu et al. (2023) |
| Building boundary | Area and perimeter | vector | 2014-2023 | Microsoft | https://wiki.openstreetmap.org/wiki/Microsoft_Building_Footprint_Data | / |
| | | | 2019-2023 | Shi et al., (2023) | https://doi.org/10.5281/zenodo.8174931 | |
| | | | / | Baidu Map | https://map.baidu.com/ | |

## 3 Methods

In this study, we estimated the height of individual buildings globally based on multisource remote-sensing datasets and vector-derived datasets (Fig. 2). First, we built a feature collection by integrating the statistical values of remote-sensing datasets and the morphological features of buildings. Second, we developed height models in the 33 subregions based on the eXtreme Gradient Boosting (XGBoost) method and assessed the model performance by ten-fold cross-validation. Third, we created a global building height map based on our estimated results. We analyzed the spatial patterns of building heights within cities



and compared our building height dataset with other existing global and regional building height products. Finally, we analyzed

the built-up infrastructure for countries and representative cities worldwide.

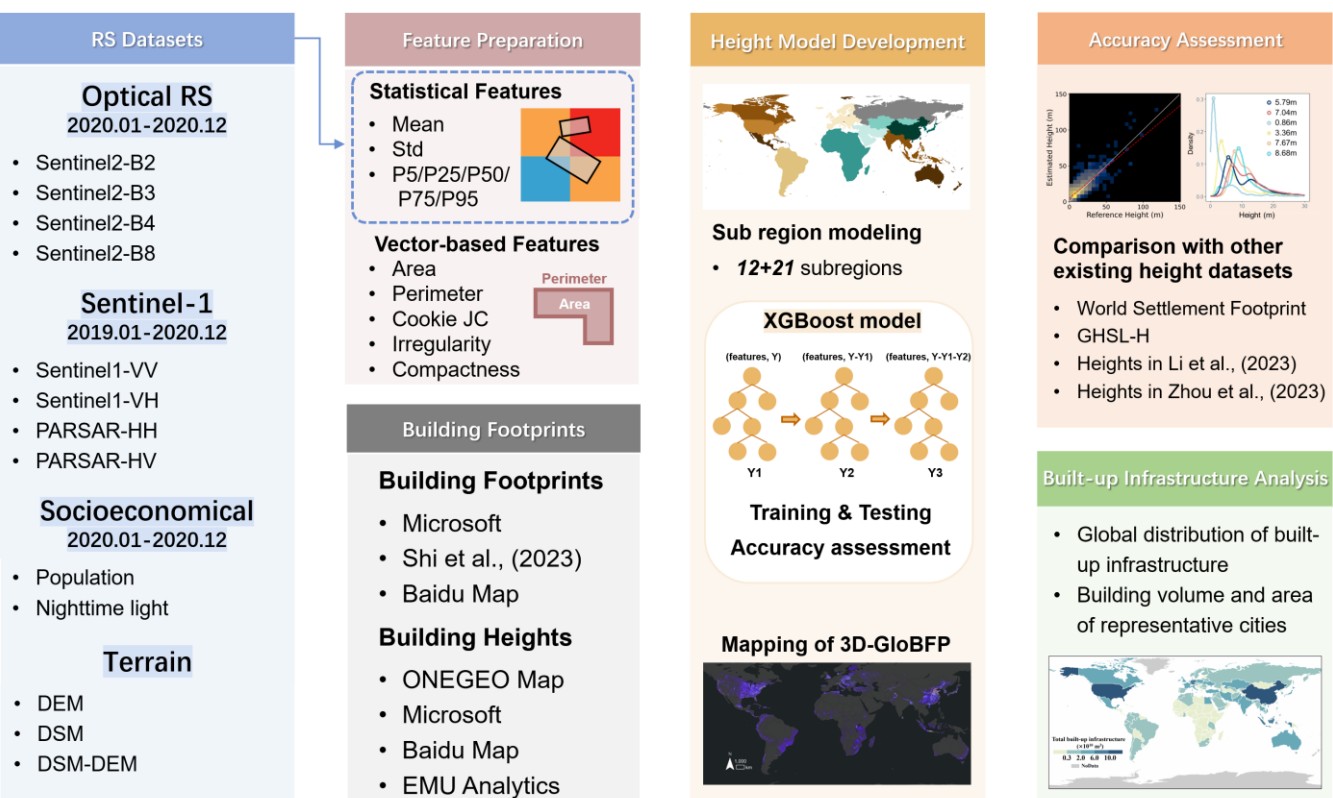

**Figure 2. Overall workflow of developing the 3D-GloBFP dataset.**

### 3.1 Feature preparation

We extracted features from multisource datasets (i.e., radar, optical, terrain, social-economic, and vector) as input features of

the models with the help of the GEE platform. First, we carried out prerecession for input remote sensing images to acquire

images with high quality. We removed pixels with a cloud percentage greater than 20 % to obtain high-quality images and

avoided the stripe effect caused by clouds. All images were reprojected to WGS84 and resampled to 10m. Second, we

aggregated remote sensing images in 2020 to vectors to get statistical information for individual buildings. Datasets from 2019

and 2021 were utilized for supplementation in areas where imagery was missing. We calculated the statistical values (i.e.,

mean value, standard deviation, and five quantiles (5 %, 25 %, 50 %, 75 %, 95 %)) of all the image pixels within each building

vector. We created fishnets of different extents with no more than 40000 buildings in each grid due to calculation memory

limitations on GEE. We exported the remote sensing image attributes for all buildings. Third, we calculated morphology

features based on building vectors, which proved effective in height estimation Arehart et al. (2021). We used five geometry

features ranging from simple (i.e., perimeter and area) to complex (i.e., compactness, fractality, and Cooke JC index) as the





input variables of the height estimation model. Compactness, fractality, and Cooke JC index were identified by building perimeter and area, measuring the complexity of the footprint of buildings (Table 2). Finally, 114 features were calculated as the input features for the height estimation model (Table S1).

**Table 2. Shape index of building footprints.**

| Feature | Notation or equation | Description |
|---|---|---|
| Compactness | $\dfrac{4\pi \times AREA\_GEO}{PERIMETER\_GEO^2}$ | Circularity or compactness of building footprint (Li et al., 2013). |
| Fractality | $1 - \dfrac{\log(AREA\_GEO)}{2 \times \log(PERIMETER\_GEO)}$ | Similarity and complexity are reflected in the relationship between area and perimeter (Basaraner and Cetinkaya, 2017). |
| Cooke JC index | $\dfrac{PERIMETER\_GEO}{4\sqrt{AREA\_GEO}} - 1$ | Shape efficiency with respect to a square (Kouskoulas and Koehn, 1974). |

## 3.2 Height model development

### 3.2.1 Division of subregions

We divided the globe into 33 regions and built the estimation models for each region, separately, considering the non-uniform spatial distribution of samples and the heterogeneity of building height distribution. Firstly, we divide the globe into 13 regions based on geographic spatial distance and regional development levels to ensure that each region has enough samples to train effective models. For instance, five Central Asian countries, including Kazakhstan, Uzbekistan, Turkmenistan, Tajikistan, and Kyrgyzstan, were aggregated together as a region for model training and estimation. However, given China's more complex urban 3D structure and more significant building heterogeneity (Wu et al., 2023), we further divided China into 21 regions. We built a separate height regression model for each region to ensure the effectiveness of the height estimation model. For instance, considering the inadequacy of samples in Northwest and Southwest China, we divided the provinces in Northwest and Southwest into one region for the training height model, respectively. We divided Beijing-Tianjin-Hebei, Yangtze River Delta, and Peral River Delta urban agglomerations into three separate areas for height model training due to the comparable economic level and population size.

### 3.2.2 Model development

We used a stratified sampling strategy to select training samples and built the height estimation model with the eXtreme Gradient Boost (XGBoost) regression method. First, we use a stratified sampling strategy to integrate the samples in each subregion. We merged all collected building height samples from each region. In each subregion, we adjusted the number of training samples in each interval according to the height distribution found in Esch et al. (2022), to ensure that the height





distribution of the sample set resembles that of each region. Then, we used the XGBoost regression model to train models in
the subregions. XGBoost is suitable for the height estimation task due to its capability to handle complex nonlinear
relationships and large-scale datasets. The number of training and testing samples was divided into 9:1. We used the Grid
SearchCV method to find the parameters (i.e., learning rate, number of estimators, max depth of trees, and lambda and alpha
in objective function). This method iterates through different parameter combinations and evaluates their performance using
cross-validation to determine the optimal model parameters. We finally built 33 XGBoost models in all subregions with
different parameters.

### 3.3 Accuracy assessment


To evaluate the height estimation models, (1) we calculated $R^2$ and RMSE in each subregion. We used ten-fold cross-validation
to assess the accuracy of the model in each region, with evaluation metrics including $R^2$ and RMSE of Ordinary Least Squares
(OLS) regression. $R^2$ evaluates the explanatory ability of variables for the dependent variable (i.e., building height), while
RMSE is used to assess the difference between estimated and reference values; (2) we compared our estimated heights with
manually measured 700 building heights in 14 cities from Google Earth Pro (Fig. 4b). The 3D measurement tool provided an
opportunity to acquire heights of individual buildings in countries around the world (Figure. 4a); (3) we evaluated the accuracy
of 3D-GloBFP and four other global datasets, with reference data collected from GIS portals of 17 cities worldwide (Table
S2). The four global height datasets include World Settlement Footprint (WSF) (Esch et al., 2022), Global Human Settlement
Layer-Built-up height (GHSL-H) (Pesaresi et al., 2021), height in Li et al. (2022), and height in Zhou et al. (2022) (Table S3);
(4) we compared the segments of 3D-GloBFP for the United States, China, and Europe with existing regional datasets (Table
S3), given the comparatively more affluent data availability within these three regions. In the US, we compared our estimated
results with two other vector-level datasets from Arehart et al. (2021) and Microsoft (2020), which cover the entire country
and have the same scale (i.e., building scale) as our datasets. The reference building heights in the US were collected from 6
city government GIS portals (Fig. 8a). In China, we validated the spatial distribution and estimation accuracy of our estimated
data against datasets from Chinese Building Height (CNBH) (Wu et al., 2023) and height in Huang et al. (2022), both of which
provide coverage for the entire country. We randomly extracted 20000 buildings from Baidu (https://ditu.baidu.com) within
Global Urban Boundaries (GUB) (Li et al., 2020c) as the reference heights (Fig. S2). Additionally, in Europe, we contrasted
the numerical distribution of building heights from our estimated data with those from WSF, height in Li et al. (2022), GHSL-
H data, and reference data. We used Urban Atlas Building Height for Europe (UABH-E)
(https://land.copernicus.eu/en/products/urban-atlas/building-height-2012) with a resolution of 10 m as the reference height,
providing building heights in core urban areas in 870 cities across Europe.

### 3.4 Built-up infrastructure analysis

We analyzed the built-up infrastructures by calculating the total building volume in countries and cities. First, we summed the
building volume for each country and created a global distribution map of built-up infrastructure across the world. To quantify



each country's contribution to the global built-up infrastructure, we calculated the proportion of each country's total building volume relative to the global total. Next, we focused on the built-up infrastructures in representative cities across various continents worldwide. We analyzed both 3D (i.e., building volume) and 2D (i.e., building area) built-up infrastructures to provide a detailed comparison. Specifically, we compared the total amounts and rankings of 3D and 2D built-up infrastructures across these cities. The boundaries of countries and cities were derived from GADM maps (https://gadm.org/). This analysis

allowed us to gain a deeper understanding of the spatial distribution characteristics and total volume features of built-up infrastructures in the world.

## 4 Results and discussion

### 4.1 Performance of the building height estimation model

The estimated building height showed consistency with reference building height across all regions in the world (Fig. 3).

Across different areas, the $R^2$ between the estimated and reference building height ranges from 0.66 (i.e., Europe) to 0.96 (South America). $R^2$ of around 40 % of regions exceeded 0.80, indicating the similarity between estimated and reference height. The RMSEs vary significantly across different areas, ranging from 1.92 m (i.e., South America) to 14.60 m (Japan, North and South Korea). 62 % of the RMSEs are less than 10 m, indicating that in most of the regions, our estimated heights are in agreement with reference heights on the building scale. The estimated heights in 5 areas are very

close to the reference height with RMSEs less than 5 m, including the United States (3.35 m), Russia (4.99 m), Med America (2.40 m), Australia (2.23 m), and South America (1.92 m). Additionally, low-rise buildings show less uncertainty compared with high-rise buildings. RMSEs of low-rise buildings (height<20 m) are generally below 6 m, especially in Western countries such as the United States (2.44 m in 0-10 m interval and 2.64 m in 10-20 m interval), and South America (1.43 m in 0-10 m interval and 4.75 m in 10-20 m interval). On the contrary, high-rise buildings

(height≥20 m) have more significant uncertainties in the estimation results. These errors may come from the saturation effect and shadow occlusion of remote sensing images in the area of high buildings (Frantz et al., 2021). It is worth noting that the uncertainty of high-rise buildings contributes significantly to regional RMSE. For instance, in Africa, the overall RMSE is 9.87 m, with high-rise buildings (i.e., ≥50 m) showing an RMSE of 25.52 m, while buildings below 10 m and in 10-20 m intervals have RMSEs of 3.86 m and 5.28 m, respectively.

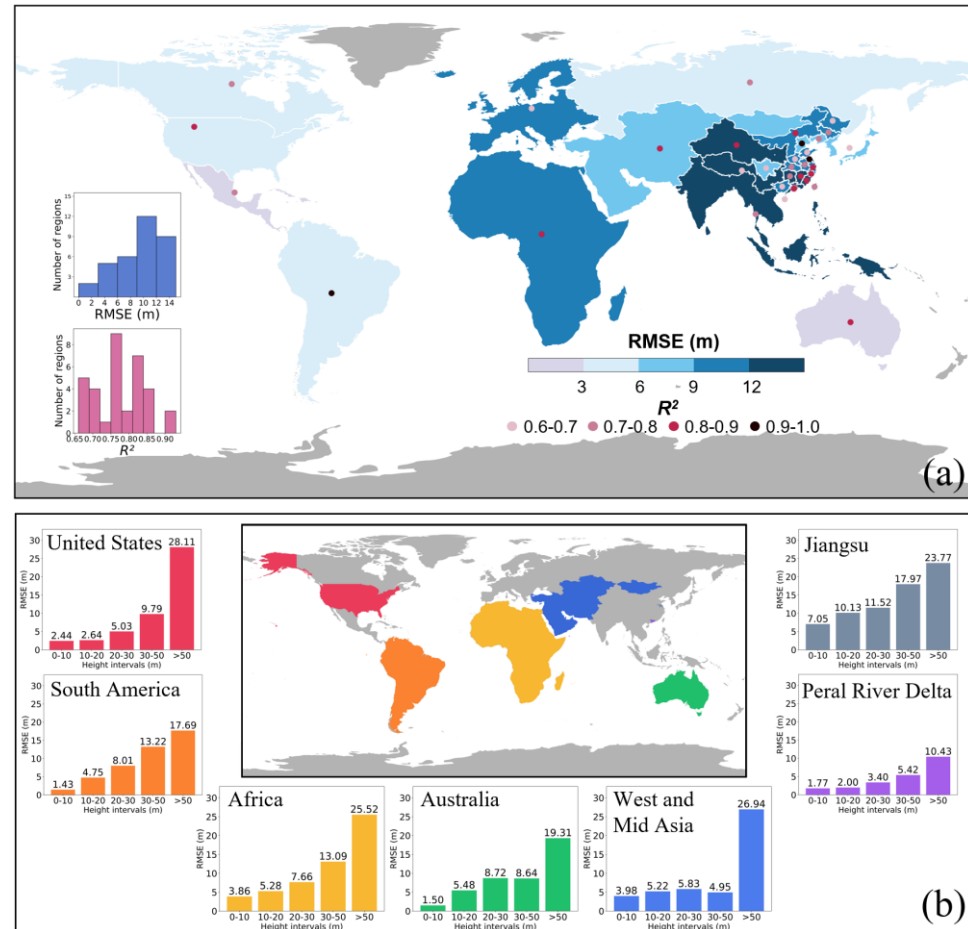


**Figure 3. Model performance in subregions.** (a) $R^2$ and RMSE of models in the subregions. (b)RMSEs of representative subregions within different height intervals.

### 4.2 Comparison with Google Earth building heights

The validation results with interpreted heights from Google Earth Street Views indicate the estimated results are consistent
with the reference heights in the metropolitans of countries around the world, particularly for those landmark buildings. We manually measured 700 buildings in 14 metropolitans across the northern and southern hemispheres (e.g., New York, London, Brasilia, and Cape Town) (Fig. 4a-b) and compared these measurements with our estimated heights. The correlation results suggest that our estimated heights show relatively high agreement with measured heights, with an $R^2$ of 0.85 and RMSE of 11.01 m. The examples of landmark buildings further confirm the effectiveness in estimating individual building heights,
especially for high-rise buildings with more considerable uncertainty as mentioned in Section 4.1 (Fig. 4d). For instance, the Federal Reserve Bank in Chicago, with a height of 113.3 m, where the difference between estimated and measured height is only 2.2 m.

**Figure 4. Comparison of estimated and interpreted heights using the 3-D Google Earth Street Views.** (a) Diagram of
measurement method in Googe Earth Pro. (b) Distribution of cities with measured building heights. (c) Overall performance
of estimated heights worldwide. (d) Measured and estimated the height of individual buildings within cities. Images in (a) and
(d) are from © Google.

## 4.3 Comparison with existing building height products

### 4.3.1 Comparison with global height products

Our estimated building heights provide more details of urban morphology and show more accurate results compared to the
other four existing global datasets (Fig. 5), including WSF (at 90 m spatial resolution), Global Human Settlement Layer: Height
(GHSL-H) (at 100 m spatial resolution) (Pesaresi et al., 2021), height in Zhou et al. (2022) (at 500 m spatial resolution), and





height in Li et al. (2022) (at 1 km spatial resolution). First, we mapped the estimated height and other four datasets and compared them to the ONEGEO Map reference height to evaluate the spatial pattern of building heights. Our estimated building height result show similar spatial patterns to the reference building heights in representative cities around the world. Specifically, the estimated heights are close to the reference height data for high-rise buildings, capturing the high-density building core of the town in the CBDs of various major cities (e.g., Downtown in Houston (Region 1), CBD of Yuexiu District in Guangzhou (Region 2), and Kowloon in Hong Kong (Region 3)). However, GHSL-H (Pesaresi et al., 2021), height in Zhou et al. (2022), and height in Li et al. (2022) can only reflect the vague spatial location of the city center, presenting various degrees of significant underestimations in the specific numerical values of building heights. The underestimation of high-rise buildings and skyscrapers is relatively substantial in GHSL-H (Pesaresi et al., 2021) and height in Li et al. (2022). Zhou et al. (2022) notably underestimate urban centers because they include nonbuilding impervious surfaces (e.g., streets and parking lots). Furthermore, compared to the WSF dataset, our estimated height can reflect a complex urban landscape with mixed high- and low-rise buildings. For instance, the spatial distribution of our derived dataset is closer to the reference dataset in Kowloon, Hong Kong, while the WSF (Esch et al., 2022) height dataset results in clusters of high-rise buildings. Additionally, our estimated heights are also more consistent with the reference datasets for low-rise building areas. For low-rise buildings within urban cores, such as the urban villages in Guangzhou (Region 4) and the low-rise structures in Tokyo's city center (Region 5), our data can provide relatively accurate numeric estimations and spatial patterns of their heights. For low-rise buildings in the surrounding area of cities, such as the southern of Santa Monica Blvd in West Hollywood, Los Angeles (Region 6), and the northwest of Geelong (Region 7), our building-scale results can reflect the morphology of these low-rise structures. However, other datasets generally show slight overestimations, especially the estimations by Li et al. (2022). For instance, building heights in northwest Geelong are below 5 m, whereas in Li et al. (2022), the building heights are between 5-10 m in that area. Besides, our estimated heights accurately showed the spatial heterogeneity of building heights between densely high-rise buildings and low-rise buildings, benefiting from a finer resolution at the scale of individual buildings. Conversely, the resolution of the other three datasets is insufficient to reflect the spatial heterogeneity of building heights due to the significant differences in building height within each pixel.

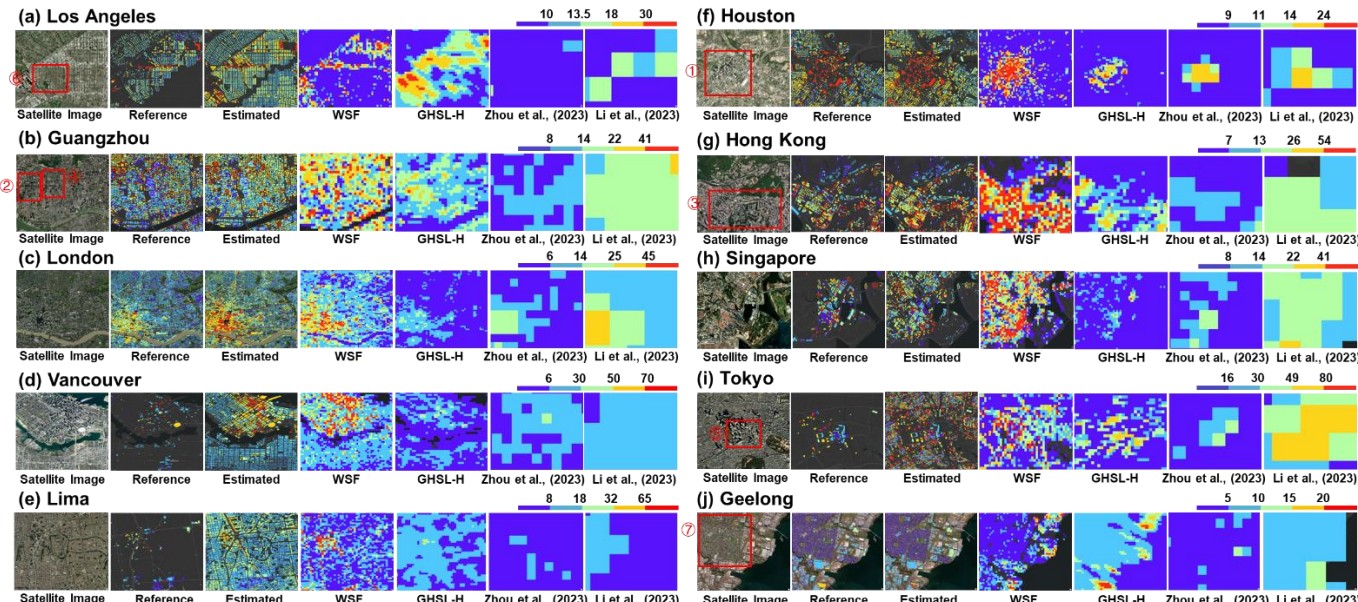

**Figure 5. Comparison of 3D-GloBFP maps with multi-scale building height products in 10 cities across the world.** (a) Los Angeles. (b) Guangzhou. (c) London. (d) Vancouver. (e) Lima. (f) Houston. (g) Hong Kong. (h) Singapore. (i) Tokyo. (j) Geelong. Note: the areas boxed represent: ① Downtown Houston. ② CBD of Yuexiu District. ③ Kowloon in Hong Kong. ④ Urban village in Guangzhou. ⑤ City center of Tokyo. ⑥ South of Santa Monica Boulevard, West Hollywood. ⑦ Northwest Geelong, respectively. The satellite images are from © Esri, © Maxar, © Earthstar Geographics, and the GIS user community.

Additionally, the height distribution and correlation results also confirm the superiority of our derived datasets in cities across the northern and southern hemispheres in the world (Fig. 6). We aggregated the high-resolution data to 1 km to match the low-resolution data, benefiting from the comprehensive coverage that densely covers the entire cities or specific urban districts. The reference heights were collected from government open-source data portals (Table S2) and were not used for training the height estimation models. Our results showed a good agreement with the reference dataset, with a peak difference of 1.25 m. Notably, 3D-GloBFP can depict the bimodal distribution of building height. In contrast, other estimation results are mostly unimodal and have some degree of underestimation (i.e., Esch et al. (2022) and Zhou et al. (2022)) or overestimation (i.e., Li et al. (2022) and Pesaresi et al. (2021)) (Fig. 6a). Moreover, the correlation results indicate that our building height dataset is consistent with the reference height, with an $R$ of 0.82 and RMSE of 6.14 m (Fig. 6b). Our estimations are closer to the reference dataset across different height intervals. However, all these datasets show a tendency to overestimate low-rise buildings and underestimate high-rise buildings. Specifically, WSF (Esch et al., 2022) dataset shows a significant overestimation of low-rise buildings, particularly those under 20 m, with an $R$ of 0.43 and RMSE of 12.40 m (Fig. 6c). GHSL-H (Pesaresi et al., 2021) and height in Zhou et al. (2022) significantly underestimated the height of high-rise buildings (>50


m), resulting in a deviation of the fitted line from the 1:1 line (Fig. 6d-e). The height dataset in Li et al. (2022) slightly underestimated high-rise buildings, but the underestimation is more severe compared to our estimations (Fig. 6f).

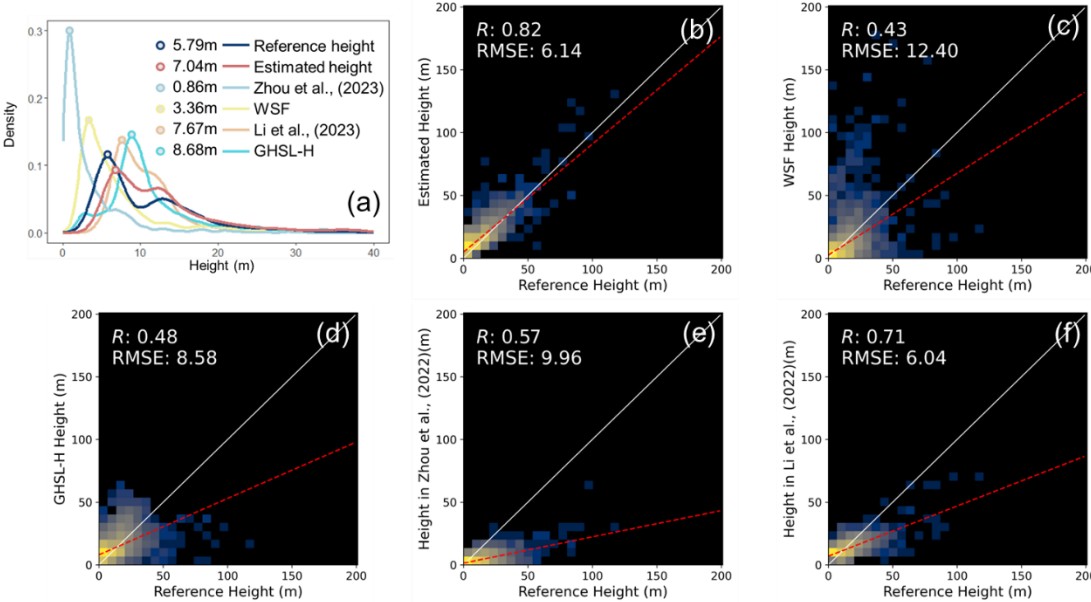

**Figure 6. Comparison of reference height, 3D-GloBFP, and other existing global products.** (a) Histogram of reference height, estimated height, and other four existing height products. (b) Scatter plot of estimated heights and reference heights. (c) Scatter plot of WSF (Esch et al., 2022) and reference heights. (d) Scatter plot of GHSL-H (Pesaresi et al., 2021) and reference heights. (e) Scatter plot of height in Zhou et al. (2022) and reference heights. (f) Scatter plot of height in Li et al. (2022) and reference heights. Note: The red dashed lines represent the regression lines fitting the reference heights against the estimated heights for each dataset. The white solid line represents the 1:1 line.

#### 4.3.2 Validation in the US, China, and Europe

We conducted additional product comparisons to evaluate our building height dataset in the United States, China, and Europe due to the relative abundance of building height products in these three regions. We initially compared the distribution of our 3D-GloBFP and other global products across three areas and subsequently compared our height dataset with other independent products within these three regions. Our 3D-GloBFP shows the most similar numerical distribution patterns to the reference heights across the United States, China, and Europe (Fig. 7). We used the building height collected from government GIS portals as the reference height. The comparison in the US indicates that our 3D-GloBFP can capture the bimodal distribution of building heights, with peaks approximately around 5 m and 12 m. Furthermore, the distribution of 3D-GloBFP in China consists of reference heights, with the peaks at 13.39 m and 16.13 m, respectively. Besides, the distribution pattern of 3D-GloBFP in Europe closely resembles the reference height despite slight overestimations. Conversely, the height in Li et al. (2022) and GHSL-H (Pesaresi et al., 2021) are generally overestimated the building heights across these three regions, while





the height in WSF (Esch et al., 2022) and results in Zhou et al. (2022) show certain underestimation compared to the reference heights.

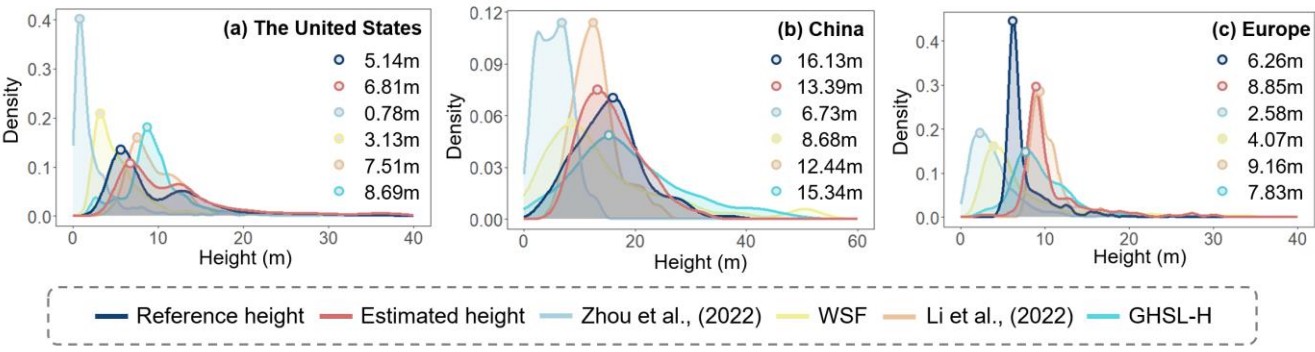

**Figure 7. Histogram of reference height, 3D-GloBFP, and other existing products in The United States (a), China (b), and Europe (c).**

Additionally, we collected several independent height datasets in these three regions and assessed the quality of 3D-GloBFP with these height datasets. First, we found that our 3D-GloBFP agreed better with the reference height than other local building scale datasets in the US. We compared our estimated building heights with two products at individual building scale in the US using building heights collected from 6 city government GIS portals as the reference height, including Boston, Louisville, New York, Boulder, Newport News, and Portland (Fig. 8a). These reference heights are independent datasets that were not used for training. Our derived results can better characterize the building heights than dataset provided by Microsoft (Microsoft, 2020) and height in Arehart et al. (2021), with an *R* of 0.68, and RMSE of 16.42 m (Fig. 8b). Overall, our estimated heights tend to underestimate building heights, especially in high-rise buildings. However, the underestimation is more evident in Microsoft building heights (Microsoft, 2020) and heights in Arehart et al. (2021), with an *R* of 0.48 and 0.38, respectively (Fig. 8c-d). The RMSE of height in Arehart et al. (2021) and reference height (i.e., 15.13 m) is slightly smaller than in our derived height dataset and reference dataset. Nevertheless, the height in Arehart et al. (2021) more significantly overestimated the height in low-rise buildings (<8 m) and underestimated the height of high-rise buildings (>40 m). It is worth noting that higher data resolution (i.e., building scale) often reveals more details of local height variations and urban landscape differences, leading to increased uncertainty.

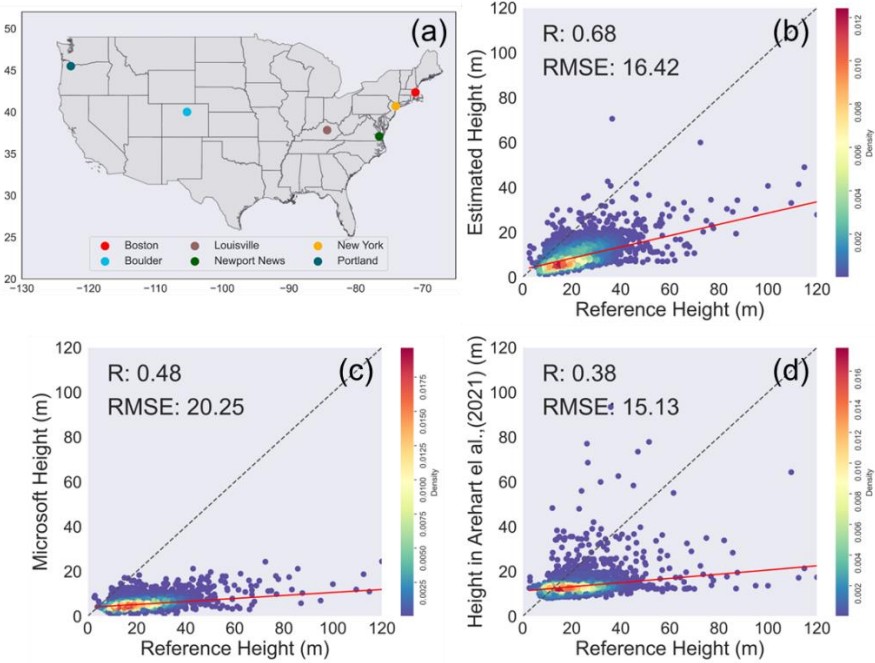

**Figure 8. Building-scale comparison to Microsoft height** (Microsoft, 2020) **dataset and height in** Arehart et al. (2021) **in the United States.** (a) Distribution of cities with reference building height. (b) Scatter plot of estimated heights and reference heights. (c) Scatter plot of Microsoft heights (Microsoft, 2020) and reference heights. (d) Scatter plot of height in Arehart et al. (2021)and reference heights.

Second, our 3D-GloBFP is similar to the reference height in terms of distribution and spatial patterns in China. We compared the numerical distribution, correlation coefficients and spatial patterns of building height with two existing building height datasets covering all the regions of China, including the CNBH height dataset (Wu et al., 2023) (at 10 m spatial resolution), heights in Huang et al. (2022) (at 30 m spatial resolution) (Fig. 9). To evaluate the estimated results in both high- and low-rise buildings within cities, we extracted 20000 random buildings in GUBs that were not used in training height estimation model as the reference height (Li et al., 2020c). The distribution results demonstrate that our 3D-GloBFP more accurately depicts the distribution of building height in China, showing superior consistency with the reference datasets across all height intervals. Conversely, CNBH (Wu et al., 2023) and Huang et al. (2022) demonstrate an overall underestimation of building heights, lacking precision in estimating the high-rise buildings in urban centers. Likewise, our derived height dataset shows the closest height values to the reference data among the three datasets, with an *R* of 0.67 and an RMSE of 13.17 m. Notably, our correlation results surpass those datasets in Huang et al. (2022) and Wu et al. (2023) datasets, with *R* of 0.32 and 0.59, respectively. Although all the uncertainties in the estimated high-rise buildings are relatively more considerable, the heights of Huang et al. (2022) and Wu et al. (2023) showed a more significant difference between estimated and reference heights. The spatial distribution maps further confirm the similarity between our estimated height and the reference height. Our height dataset can capture the spatial distribution and values of high-rise buildings, including landmarks such as the Lujiazui Financial

and Trade Zone in Shanghai (Region 1) and the CBD in Chaoyang District in Beijing (Region 2). In contrast, CNBH (Wu et al., 2023) notably underestimates heights in the CBD areas. While height in Huang et al. (2022) approximates the spatial patterns in Beijing, it significantly underestimates clustered high-rise buildings in the Lujiazui Financial and Trade Zone in Shanghai. Furthermore, our height dataset can identify the low-rise residential buildings of old urban areas (e.g., buildings near Tongfu Middle Road in Guangzhou (Region 3)). Conversely, CNBH (Wu et al., 2023) overestimates the heights of low-rise buildings in old urban areas. The results in Huang et al. (2022) are similar to the reference height in old urban areas. However, the results of Huang et al. (2022) misidentified contiguous taller buildings (20-36 m) around old urban areas as high-rise buildings (>36 m), which may contribute to resolution limitations, resulting in insufficient recognition of height heterogeneity within complex urban landscapes.

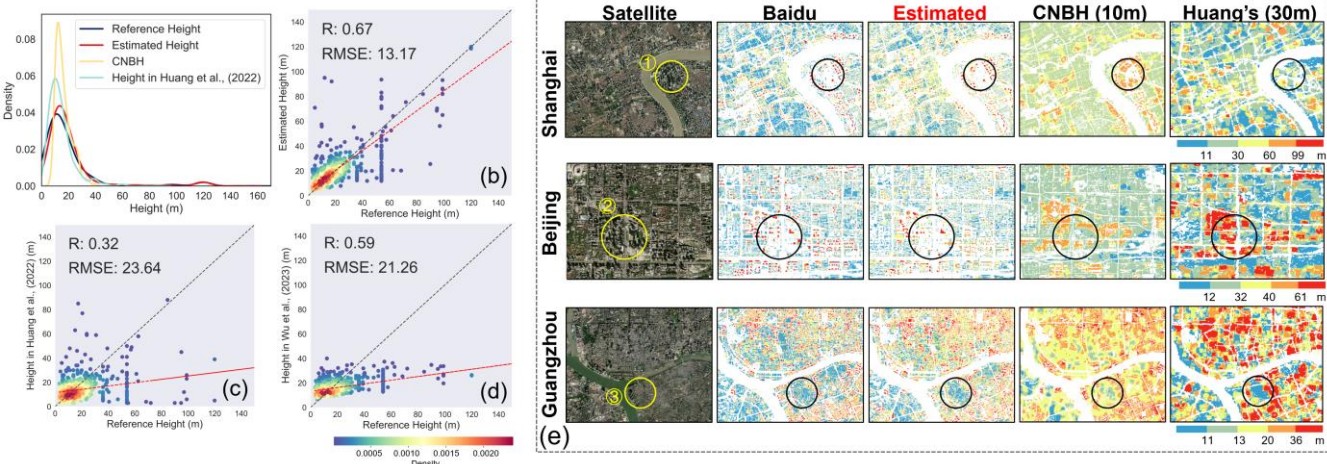

**Figure 9. Comparison of height in China by** Huang et al. (2022) **and** Wu et al. (2023)**.** (a) Distribution of test points in GUBs. (b) Scatter plot of estimated heights and reference heights. (c) Scatter plot of height in Huang et al. (2022) and reference heights. (d) Scatter plot of height in Wu et al. (2023) and reference heights. (e) Spatial patterns of building height in Shanghai, Beijing, and Guangzhou. Note: the areas boxed represent: ① Lujiazui Financial and Trade Zone. ② CBD in Chaoyang District. ③ Community near Tongfu Middle Road, respectively. The satellite images are from © Esri, © Maxar, © Earthstar Geographics, and the GIS user community.

Additionally, the numerical distribution of 3D-GloBFP is more consistent with the reference height than the other three products in Europe (Fig. 10). We quantified the differences between 3D-GloBFP, reference heights, and three other datasets with histograms. We aggregated UABH-E data covering 870 cities in Europe to a 1 km resolution as reference data. The distribution of 3D-GloBFP closely resembled that of the reference data, with similar peak values. The reference data shows the highest frequency of building heights in the range of 2.5-5 m, while the estimated data indicates the highest frequency of building heights in the range of 5-7.5 m. However, we observed an overestimation of low-rise buildings of 3D-GloBFP in




Europe. Moreover, height in Li et al. (2022) and GHSL-H (Pesaresi et al., 2021) show more obvious overestimations. In contrast, WSF (Esch et al., 2022) underestimate the buildings with heights larger than approximately 5 m.

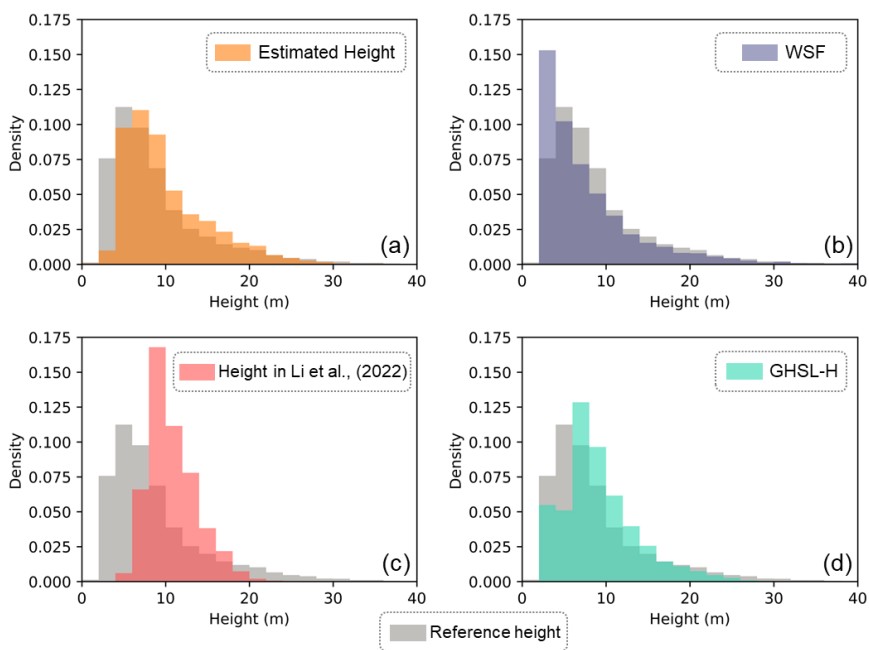


**Figure 10. Distribution of building height in Europe.** (a) 3D-GloBFP. (b) WSF (Esch et al., 2022). (c) Height in Li et al. (2022). (d) GHSL-H (Pesaresi et al., 2021).

## 4.4 Mapping of global building height

The global building height exhibits a distinct spatial pattern across regions, countries, and within cities (Fig. 11). Our global-
coverage height maps indicate that low-rise buildings dominate globally, while high-rise buildings are dispersed. Low-rise buildings are commonly found in urban centers and outskirts across countries and regions, while high-rise buildings are predominantly concentrated in relatively developed areas within cities. The building height map suggests a noticeable surface roughness of the built-up environment globally. For instance, in developed regions like eastern China and the eastern United States, there are more high-rise buildings. Meanwhile, in developing regions such as Sub-Africa, building heights are
comparatively lower. Our building-scale height maps reveal significant height heterogeneity within the cities. Specifically, high-rise buildings are generally located in the commercial land of urban centers, with building heights gradually decreasing from the city center to the surrounding rural areas in a radial pattern.

Earth System
Science
Data

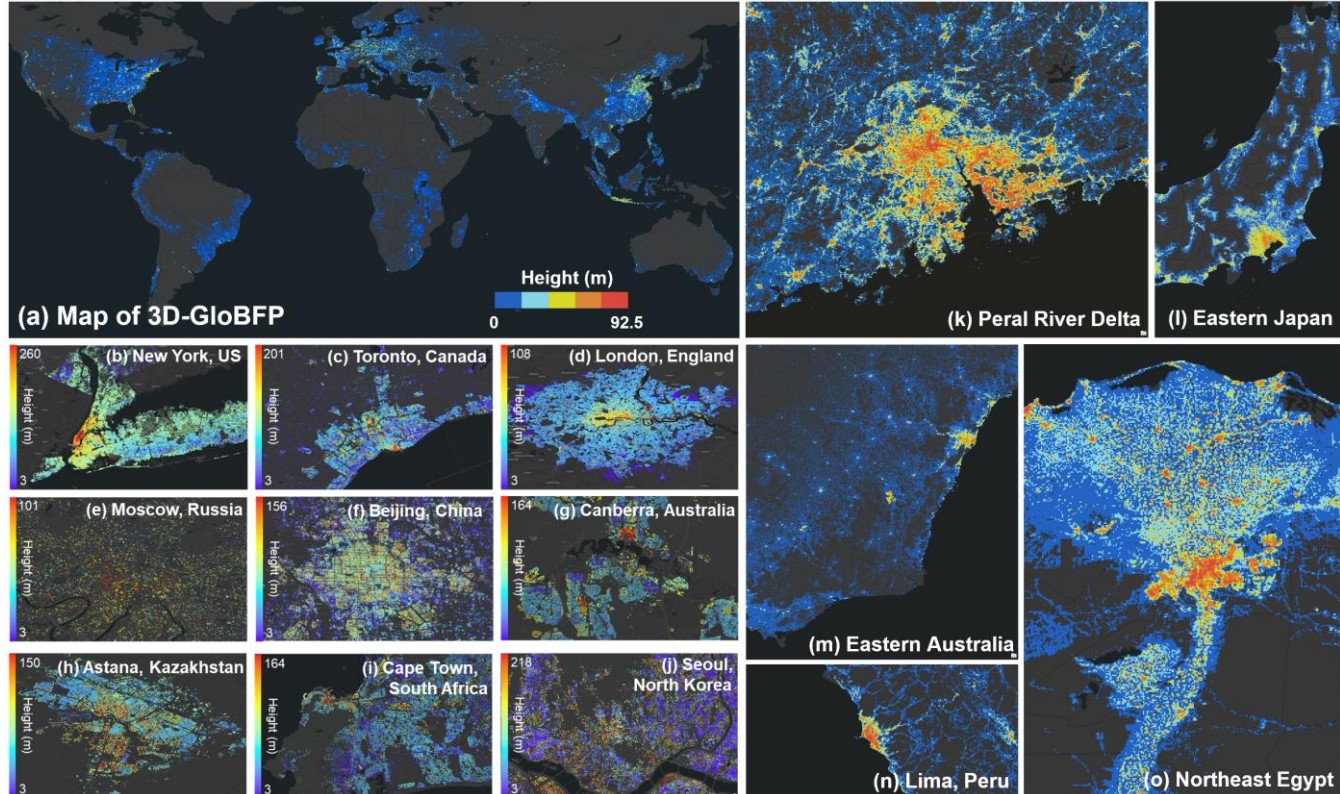

**Figure 11. Spatial variations of building heights in the world.** (a) Map of 3D-GloBFP. (b-j) Large view of representative
cities in the world at building scale. (k-o) Large view of representative regions in the world at 1km scale. Note: Colorbars of
k-o are the same as that in (a).

### 4.5 Global disparities in built-up infrastructure

### 4.5.1 Global distribution of built-up infrastructure

Our findings reveal a notably uneven distribution of built-up infrastructure across different countries globally. We calculated
the total built-up infrastructure (i.e., a sum of building volume) (Fig. 12a). We determined its global proportion for each country
based on 3D-GloBFP (Fig. 12b). We found that developed nations and certain rapidly emerging economies show a more
significant proportion of the total volume of built-up infrastructures. In contrast, countries and regions with lower levels of
economic development hold relatively lower volumes of built-up infrastructures. The built-up infrastructures in China, the US,
and several European countries significantly surpass that of other regions, contributing the majority of the global built-up
infrastructure. Specifically, China is the country with the largest total built-up infrastructure volume globally $(5.28 \times 10^{11}$ m$^3$,
accounting for 23.9 % of the global total), followed by the United States $(3.90 \times 10^{11}$ m$^3$, accounting for 17.6 % of the global
total). Other countries with significant infrastructure volumes include Germany $(9.39 \times 10^{10}$ m$^3$, accounting for 4.2 % of the



global total), Indonesia ($6.62\times10^{10}$ m$^3$, accounting for 3.0 % of the global total), and France ($5.66\times10^{10}$ m$^3$, accounting for 2.5 % of the global total). The total volume of built infrastructure in Africa is relatively low, accounting for a small percentage

of the global total (e.g., Angola ($2.53\times10^9$ m$^3$, accounting for 0.11 % of the global total), Zimbabwe ($2.10\times10^9$ m$^3$, accounting for 0.09 % of the global total), Tanzania ($3.99\times10^9$ m$^3$, accounting for 0.18 % of the global total)).

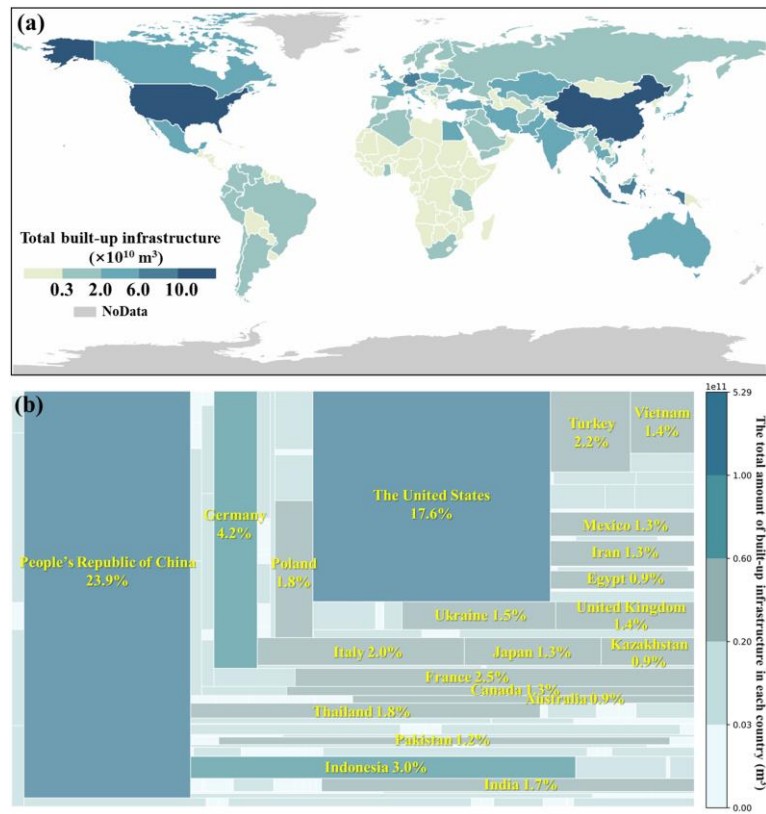

**Figure 12. Built-up infrastructures in the world.** (a) Total built-up infrastructures in each country. (b) The shares of built-up infrastructures in each country.

**4.5.2 Comparison in building volume and area of representative cities**

The building volume and area of representative cities vary significantly across different regions worldwide. The disparity in building volume across cities is pronounced. For instance, Shanghai, China ($2.06\times10^{10}$ m$^3$) exhibits a building volume approximately 21 times larger than that in Pyongyang, North Korea ($9.85\times10^8$ m$^3$). We found that Chinese representative cities with building volume exceed that of representative cities elsewhere in the world due to their higher population density and

larger administrative divisions. It is worth noting that while the building area of Beijing ($9.76\times10^8$ m$^2$) surpasses that in Shanghai ($8.49\times10^8$ m$^2$), the building volume in Shanghai ($2.1\times10^{10}$ m$^3$) is more significant than Beijing ($1.3\times10^{10}$ m$^3$) due to

its more efficient utilization of vertical urban space, resulting in higher average building heights of 16.7 m in Shanghai compared to 10.0 m in Beijing. In North America, the sum of building areas is similar in representative cities, but New York City has significantly larger building volumes ($6.99 \times 10^9$ m$^3$). This disparity can be attributed to the limited and expensive land

resources in New York, which promotes the city's adoption of vertical development strategies, particularly in Manhattan, where numerous high-rise buildings are concentrated. Despite having the most extensive building area ($7.06 \times 10^9$ m$^2$) among European representative cities, London's overall volume is lower ($7.06 \times 10^9$ m$^3$) due to its lower average height, influenced by the abundance of low-rise and historical buildings that occupy significant space within the urban landscape. In contrast, the building volume of representative cities in South America, Africa, and Australia are generally small (e.g., Brazillia, Brazil,

with $2.70 \times 10^8$ m$^3$ building volume, Cape Town, South Africa, with $1.48 \times 10^9$ m$^3$ building volume, Sydney, Australia, with $3.3 \times 10^8$ m$^3$ building volume).

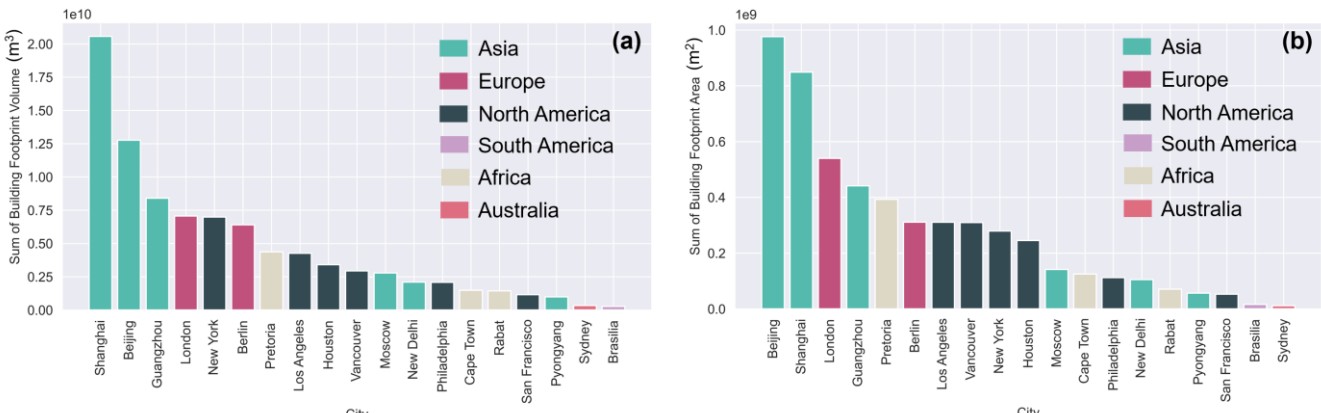

**Figure 13. Building volume and area in representative cities in the world.** (a) The sum of the building footprint volume.
(b) The sum of the building footprint area.

**5 Data availability**

The 3D-GloBFP dataset is available at https://doi.org/10.5281/zenodo.11319913 (Building height of the Americas, Africa, and Oceania in 3D-GloBFP) (Che et al., 2024a), https://doi.org/10.5281/zenodo.11397015 (Building height of Asia in 3D-GloBFP) (Che et al., 2024b), and https://doi.org/10.5281/zenodo.11391077 (Building height of Europe in 3D-GloBFP) (Che et al.,
2024c). The dataset is stored in shapefile format with building height in the attribute table.



## 6 Conclusions

In this study, we have released a global building height dataset at the individual building scale, providing detailed building footprint information along with heights. Initially, we developed 31 height estimation models based on integrated multisource remote sensing and building morphology features. Next, we assessed the model performance and the dataset quality by cross-validation with other existing national and regional building height datasets. Our results showed that the derived height dataset has a high agreement with reference data in regions worldwide, with the models' $R^2$ ranging from 0.66 to 0.96 and RMSEs ranging from 1.9 m to 14.6 m. Moreover, estimated results are consistent with the measured height in Google Earth Street Views with an $R^2$ of 0.85. Our estimated heights also show numerical distribution and spatial patterns that are more similar to the reference heights than other existing datasets. Then, we provided a seamless building height map globally. The detailed building height map reveals the distinct landscape heterogeneity within global cities. We also found significant variations in building volume and area within cities across representative cities in each continent due to different development patterns and stages. Finally, we analyzed the built-up infrastructures in countries and cities by summarizing the total building volume. The results reveal a significant variation in built-up infrastructure distribution across countries, with developed nations and certain emerging economies holding a larger proportion. Furthermore, substantial disparities in both 3D and 2D built-up infrastructures are evident across representative cities worldwide, influenced by factors such as different development stages and patterns.

The 3D-GloBFP map is the first individual building height dataset to depict the most detailed building three-dimensional morphology worldwide, offering great potential to support studies ranging from macro-scale global analyses to micro-scale investigations within urban areas. Our developed dataset can serve as the base input for studies, such as climate modeling (He et al., 2019), population simulation (Zhao et al., 2021), building function classification (Zheng et al., 2024), and disaster assessments (Hossain and Meng, 2020). Moreover, our dataset also contributes to studies on the interaction between human society and ecosystems (Zhong et al., 2021; Rodriguez Mendez et al., 2024; Güneralp et al., 2017; Arehart et al., 2022), such as Urban Heat Island (UHI) assessment (Li et al., 2020d), carbon footprint accounting (Li et al., 2024a), and building stock analysis (Frantz et al., 2023). These studies can further contribute to addressing environmental issues related to anthropogenic activities, thereby promoting the achievement of sustainable development.



**Supplement**

Supplement includes 3 tables and 2 figures.

**Author contributions**

X.Liu and X.Li designed the research. YC and X.Li designed and performed the experiments, and wrote the original manuscript.
510 YC and YW organized the dataset. QS and JZ provided data. All authors reviewed and revised the manuscript.

**Competing interests**

At least one of the (co-)authors is a member of the editorial board of Earth System Science Data.

**Disclaimer**

**Acknowledgements**

515 We gratefully acknowledge the creation and provision to the Building Footprint and Height dataset by Microsoft, the Building Height dataset by Baidu Map, the ONEGEO Map by ONEGEO GmbH, and the Emu Analytics Building Height dataset by Emu Analytics.

**Financial support**

This research was supported by the National Science Fund for Distinguished Young Scholars (grant no. 42225107).

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
