# Peer review of "3D-GloBFP: the first global three-dimensional building footprint dataset"

_Earth System Science Data, 2024_

## Author Comment (AC1)

**Reviewer #1 (Response to Reviewer)**

**General comments**

This article uses a practical method based on multimodal data to construct the first global scale 3D information dataset of buildings. The data set provided by this study fills the gap in fine-grained building height data globally, which is of great significance for urban morphology research and climate change analysis. The model validation results are comprehensive and promising; however, a more detailed explanation of the technical methods would enhance the paper's clarity (see specific comments). Overall, the paper is well-structured, and the dataset is valuable to urban studies.

**Response**: thank you very much for your helpful comments and suggestions. We have carefully revised our manuscript and provided a point-by-point response below.

**Specific comments**

**Comment #1:** The article uses multiple sources of data for analysis, and it is recommended to add ablation experiments between different data to demonstrate the effectiveness of using the data.

**Response**: thank you for your question. We evaluate the contributions of features in different categories with added experiments.

We evaluated the impacts of input features using the US dataset to further indicate the efficiency of the synthetic-used multi-source datasets. We used single category (i.e., radar-only, optical-only, terrain-only) and different combinations of features as the model input to estimate the height of buildings (**Table R1**). According to the results, the $R^2$ and RMSE of radar-only model is 0.56 and 5.24m. The optical features provide effective information of building height estimation, with the model's $R^2$ increased 0.1 and RMSE decreased 0.7m compared with the radar-only model. The terrain features significantly increase the accuracy of estimation model, with model's $R^2$ increased 0.1 and RMSE decreased 0.8m compared with model2. Additionally, socioeconomic and vector features slightly improve the ability of model, with $R^2$ increased 0.05 and RMSE decreased 0.35m compared with model3. These results demonstrate that the synthetic use of these features is practicable and effective for building the height estimation model.

**Table R1. Performance of models with different feature combinations.**

| | Feature combinations | $R^2$ | RMSE (m) |
|---|---|---|---|
| Model1: radar-only | ①② | 0.56 | 5.24 |
| Model2: radar + optical | ①②③ | 0.67 | 4.57 |
| Model3: radar + optical + terrain | ①②③④⑤⑥ | 0.77 | 3.76 |
| Model4: radar + optical + terrain + socioeconomic | ①②③④⑤⑥⑦⑧ | 0.79 | 3.65 |
| Model5: radar + optical + terrain + socioeconomic + vector | ①②③④⑤⑥⑦⑧⑨ | 0.82 | 3.39 |

① Sentinel1_VV_mean, Sentinel1_VV_std, Sentinel1_VV_percentile5, Sentinel1_VV_percentile25, Sentinel1_VV_percentile50, Sentinel1_VV_percentile75, Sentinel1_VV_percentile95, Sentinel1_VH_mean, Sentinel1_VH_std, Sentinel1_VH_percentile5, Sentinel1_VH_percentile25, Sentinel1_VH_percentile50, Sentinel1_VH_percentile75, Sentinel1_VH_percentile95

② PALSAR_HH_mean, PALSAR_HH_std, PALSAR_HH_percentile5, PALSAR_HH_percentile25, PALSAR_HH_percentile50, PALSAR_HH_percentile75, PALSAR_HH_percentile95, PALSAR_HV_mean, PALSAR_HV_std, PALSAR_HV_percentile5, PALSAR_HV_percentile25, PALSAR_HV_percentile50,

PALSAR_HV_percentile75, PALSAR_HV_percentile95

③ Sentinel2_Band2_mean, Sentinel2_Band2_std, Sentinel2_Band2_percentile5, Sentinel2_Band2_percentile25, Sentinel2_Band2_percentile50, Sentinel2_Band2_percentile75, Sentinel2_Band2_percentile95, Sentinel2_Band3_mean, Sentinel2_Band3_std, Sentinel2_Band3_percentile5, Sentinel2_Band3_percentile25, Sentinel2_Band3_percentile50, Sentinel2_Band3_percentile75, Sentinel2_Band3_percentile95, Sentinel2_Band4_mean, Sentinel2_Band4_std, Sentinel2_Band4_percentile5, Sentinel2_Band4_percentile25, Sentinel2_Band4_percentile50, Sentinel2_Band4_percentile75, Sentinel2_Band4_percentile95, Sentinel2_Band8_mean, Sentinel2_Band8_std, Sentinel2_Band8_percentile5, Sentinel2_Band8_percentile25, Sentinel2_Band8_percentile50, Sentinel2_Band8_percentile75, Sentinel2_Band8_percentile95

④ DEM_mean, DEM_std, DEM_percentile5, DEM_percentile25, DEM_percentile50, DEM_percentile75, DEM_percentile95

⑤ DSM_mean, DSM_std, DSM_percentile5, DSM_percentile25, DSM_percentile50, DSM_percentile75, DSM_percentile95

⑥ nDSM_mean, nDSM_std, nnDSM_percentile5, nDSM_percentile25, nDSM_percentile50, nDSM_percentile75, nDSM_percentile95

⑦ population_mean, population_std, population_percentile5, population_percentile25, population_percentile50, population_percentile75, population_percentile95

⑧ nighttimelight_mean, nighttimelight_std, nighttimelight_percentile5, nighttimelight_percentile25, nighttimelight_percentile50, nighttimelight_percentile75, nighttimelight_percentile95

⑨ building area, building perimeter

**Comment #2:** Why did you choose to use XGboost instead of random forest or support vector machine? Please provide additional experiments or explanations.

**Response**: thank you for your question. We have tested the performance of machine learning models in preliminary experiments, and finally used eXtreme Gradient Boosting (XGB) model due to its accuracy and efficiency (**Table R2**). Specifically, we compared the model performance of Decision Tree (DT), Random Forest (RF), Gradient Boosting Regression (GB), XGB, and Support Vector Machine (SVM), using 135291 training samples and 15033 testing samples in Africa. The results showed that the XGB model has the highest accuracy (testing $R^2 = 0.733$, RMSE = 5.213m). The performance of GB model is slightly inferior to that of the XGB model with testing $R^2$ of 0.731 and RMSE of 5.223m. The results of RF model showed overall underestimation with lower accuracy (testing $R^2 = 0.688$, RMSE = 5.625m). And the results of DT model show significant overestimation for building heights below 50m. The SVM model is ineffective to estimate height of buildings. These experiments were conducted using Intel® Core™ i9 with Python 3.9.

**Table R2. Performance of models with different feature combinations**.

| Model | Model parameters | Training $R^2$ | Training RMSE (m) | Testing $R^2$ | Testing RMSE (m) | Training time | Scatter plot |
|---|---|---|---|---|---|---|---|
| DT | 'max_depth': 10, 'min_samples_split': 2, 'min_samples_leaf': 1 | 0.723 | 5.643 | 0.519 | 6.988 | 10s |  |
| RF | 'n_estimators': 1000, 'max_depth': 10, 'min_samples_split': 2, 'min_samples_leaf': 1, 'max_features': 'sqrt', 'bootstrap': True | 0.758 | 5.266 | 0.688 | 5.625 | 514s |  |
| GB | 'n_estimators': 1000, 'max_depth': 10, 'min_samples_split': 2, 'min_samples_leaf': 1, 'learning_rate': 0.1, 'subsample': 0.8, 'max_features': 'sqrt' | 0.919 | 3.041 | 0.731 | 5.223 | 67s |  |
| **XGB** | 'max_depth': 10, 'eta': 0.1," 'lambda': 30, 'alpha': 20, 'colsample_bytree':0.7, 'learning_rate' : 0.1, 'n_estimators': 1000 | **0.986** | **1.234** | **0.733** | **5.213** | **46s** |  |
| SVM | 'kernel': 'rbf', 'C': 1.0, 'epsilon': 0.1 | 0.190 | 9.064 | 0.169 | 9.768 | 3179s | / |

**Comment #3:** Please explain the specific operation of manual measurement in section 4.2 and the basis for the authenticity of manual height measurement values.

**Response**: thank you for your comment. We used manually measured building heights on Google Earth Pro as the reference height to evaluate the quality of our height results. We measured the length of the ridge lines of the 3D building models using a 3D measurement tool as the height of individual building (**Fig. R1**). The vertical information of these 3D building models is obtained using Google Street View, aerial photographs, and satellite images.

The selecting points in manual measurements can affect the precision of measured height results, especially for buildings with complex shapes. To minimize the error, we measured the smooth ridge lines of buildings. We tested the stability of the measurement result by repeatedly measuring the building heights five times. The measurement results of a 130m building fluctuate within 1m, indicating the measurement error is relatively small. Also, the measurement results of a 12.8m building fluctuate within 0.1m. This relative measurement error is less than 1%, suggesting the method is reliable to obtain reference heights.

[Figure]

**Figure R1.** Results of measuring the same building five times using the Google 3D measurement tool.

**Comment #4:** What are the advantages of building scale 3D data over course scale 3D data set?

**Response**: thank you for your question. Building-scale 3D data enhances surface morphology with finest-scale details, which can support a wider range of urban studies compared to coarse-scale building height datasets.

Building-scale 3D data refines urban analysis and simulation by offering precise height information for building structures. The vector data scale avoids confusion with other surface objects, such as impervious surfaces (e.g., roads and parking lots). This allows for more detailed urban morphology analysis and simulations (e.g., digital twin) compared to coarse-scale height datasets, including building stock calculation, building carbon emission analysis, fine-scale population simulations, and thermal environment analysis. Moreover, building-scale 3D dataset can support a broader range of fine-scale analyses when combined with increasingly high-resolution datasets. As the resolution of remote sensing products have advanced, high-resolution datasets are now available, such as 1m land cover and 1m tree height datasets. Building-scale datasets can support urban analysis at these high-resolution levels. For example, building-scale 3D datasets and tree height dataset can be used to analyzed the cooling effects of shadows from buildings and trees in urban area *(Tolan et al., 2023)*. Additionally, building-scale dataset can also aggerated to other resolutions for analysis under difference scales. Relevant contents can be found in our manuscript as:

[revised manuscript text omitted]

*2023.*

**Comment #5:** Please explain in detail how to aggregate building scale height data to coarse resolution scales for validation.

**Response**: thank you for your question. We aggregated the building-scale height dataset to 1km-resolution to compare with other existing coarse-resolution products according to Eq1:

$$\bar{H} = \frac{\sum_{i=1}^{N} H_i}{N} \tag{1}$$

where $\bar{H}$ is the aggregated results in 1km pixel, $N$ is the number of buildings in 1km pixel, and $H_i$ is the height of buildings in the 1km pixel.

Specifically, we calculated the average height of all buildings located within each grid cell. In this process, each 1km pixel represents the height of buildings with other built-up regions excluded (e.g., roads and parking lots) in the computation. Details can be found in our manuscript as:

*"We also aggregated the high-resolution data at 1 km resolution to align with the low-resolution data by calculating the average height of all buildings located within each grid cell. This approach allows us to compare the differences with the reference data at a consistent resolution across all datasets." (page 10, line 220-222)*

**Comment #6:** What are the contributions of statistical values in the model?

**Response**: thank you for raising this concern. The statistical values can improve the accuracy of the models, especially for high-rise buildings. We compared the RMSEs for models trained with mean values alone versus models trained with all statistical values (i.e., standard deviation, 5%, 25%, 50%, 75%, and 95% quantiles) of pixels intersecting with each building boundary (**Fig. R2**). We evaluated the contributions of statistical values using RMSEs across different height intervals (i.e., <10m, 10-20m, 20-30m, 30-50m, 50-100m, >100m). The results show that the statistical values slightly enhance the accuracy of estimating results in low-rise buildings and significantly increased the accuracy of high-rise buildings. For instance, the model using statistical values reduce the RMSE by 3m in estimating buildings with height of 30-50m. Furthermore, these quantiles reduce the RMSE by approximately 6m and 7m in estimating buildings with height of 50-100m, and >100m. The statistical values can reflect the complex variations of pixel values within building boundaries of high-rise buildings, providing the model with comprehensive information and enhanced its accuracy. However, the statistical value does not significantly increase the model accuracy for low-rise buildings (i.e., buildings lower than 20m), due to the similar values of pixels within buildings.

[Figure]

**Figure R2.** RMSEs of models with and without quantiles as input features.

---

## Author Comment (AC2)

**Reviewer #2 (Response to Reviewer)**

This paper presents the world's first three-dimensional building footprint dataset, 3D-GloBFP, which integrates multi-source remote sensing data and various reference building height data. By employing machine learning methods, it generates high-precision global building height data. This dataset holds significant importance across multiple application domains, including urban planning, environmental monitoring, disaster management, and energy consumption analysis. The research demonstrates substantial promise and value, providing a crucial foundation for the acquisition and application of global 3D building data. I believe that 3D-GloBFP is an indispensable foundational dataset for urban research. During the review process, I identified several areas that require further clarification and improvement.

**Response**: thank you very much for your constructive comments. We have carefully revised our manuscript and provided further responses for your further review.

**Comment #1:** In Section 3.2.1 "Division of Subregions," the information of training and testing samples (e.g., the total amount) for each sub-region should be explicitly provided. Clearly specifying the selection criteria and distribution of these samples will help readers better understand the process of model training and validation.

**Response**: thank you for pointing out this issue. We enhanced the description of our training and testing datasets, including adding the information of distribution and the number of samples in each subregion:

*"We divided the globe into 33 regions and developed the building height estimation model for each region, considering the non-uniform spatial distribution of samples and the heterogeneous building heights. Firstly, we divided the globe into 13 regions based on geographic spatial distance and regional development levels to ensure that each region has enough samples to train effective models. For instance, the Central and West Asian countries were considered as a single region for model training and estimation with 40040 training samples. However, given China's complex urban 3D structure and significant building heterogeneity (Wu et al., 2023), we further divided China into 21 regions. We built a separate height regression model for each region to ensure the effectiveness of the height estimation. For instance, considering the inadequacy of samples in Northwest China, we considered the provinces in Northwest as a single region with 8050 training samples for model training. Additionally, we considered the Beijing-Tianjin-Hebei, Yangtze River Delta, and Peral River Delta urban agglomerations as three separate regions due to the comparable economic levels and population size."* (page 8-9, line 186-195)

[Figure]

*Figure 3. Distribution of subregions.*

*Reference:*

*Wu, W.-B., Ma, J., Banzhaf, E., Meadows, M. E., Yu, Z.-W., Guo, F.-X., Sengupta, D., Cai, X.-X., and Zhao, B.: A first Chinese building height estimate at 10 m resolution (CNBH-10 m) using multi-source earth observations and machine learning, Remote Sensing of Environment, 291, 113578, https://doi.org/10.1016/j.rse.2023.113578, 2023.*

**Comment #2:** I noticed that there are some missing tiles in Ghana and incomplete regions in Guangdong, please ensure that the dataset is complete globally.

**Response**: thank you for raising this concern. We completed the building footprints with height attributes of our datasets. The updated results of Ghana and Guangdong are showed in Fig R3 and Fig R4.

[Figure]

**Figure R3.** 3D-GloBFP in Ghana.

[Figure]

**Figure R4.** 3D-GloBFP in Guangdong.

**Comment #3:** In China, reference building heights used as training samples were mostly concentrated in city centers. Please clarify the accuracy of the model's estimates for building heights in different urban areas (e.g., urban fringe). This is crucial for validating the model's applicability in diverse environments.

**Response**: thank you for the great question. We evaluated the model performance across different urban regions, including city center, suburban area, and urban fringe. The results show that the estimated results are consistent with reference data in various city environments.

To assess the model performance in different regions, we divided the cities into several rings based on road network, and calculated the $R^2$ and RMSE between estimated and reference height. It is worthy to note that these reference samples were not used for model training. We selected Shanghai, Guangzhou-Foshan, and Chengdu as samples cities due to their heterogeneity of building heights from city center to fringe areas.

The model shows relatively high reliability in estimating building heights across different rings(**Fig. R5**). In Shanghai, the model performs well in the Middle Ring ($R^2$ = 0.88, RMSE = 8.91 m), indicating the model can capture the height distributions in city center. The $R^2$ of suburban and outer ring areas is 0.90 with an RMSE of approximately 13.7 m. Although the RMSE is relatively higher in suburban ring and outer areas, the estimated results generally agree well with reference height in different regions. In Guangzhou-Foshan metropolitan area, the model's accuracy of the areas outside outer ring ($R^2$ = 0.73, RMSE = 9.51 m) is slightly higher than in the Middle Ring ($R^2$ = 0.66, RMSE = 14.02 m). The results indicate that the model performs well across different regions of Guangzhou-Foshan, with more accurate estimates in outer ring and outside areas. In Chengdu, the model's performance in the Middle Ring ($R^2$ = 0.40, RMSE = 9.41 m) is slightly lower than in the outside outer ring area ($R^2$ = 0.74, RMSE = 7.77 m), suggesting higher accuracy in the more peripheral areas. Buildings in city centers have more complex functions and higher height heterogeneity compared to suburbs, which may make it challenging to accurately estimate building heights accurately. Overall, these results demonstrate that the model performs effectively across different urban regions, providing supports for the model's application in various urban environments.

[Figure]

**Figure R5.** Model performance from the urban core to fringe areas. (a) Shanghai, (b) Guangzhou-Foshan, (c) Chengdu.

**Comment #4:** For high-rise buildings, especially super tall buildings, what are the potential reasons for height underestimation? It is recommended to include a detailed error analysis and explanation.

**Response**: thank you for your question. We discussed the reason about the underestimation of high-rise buildings in Section 4.1 as:

*"The resolution of coarse-resolution remote sensing dataset (e.g., DSM with a 30 m resolution and nighttime light with a 463.83 m resolution) make it difficult to capture the heterogeneity features of super tall buildings, especially in densely built urban cores. Moreover, height and material of high-rise buildings, as well as the side-looking scene illumination Sentinel sensor, can cause complex multipath effects, complicating radar signal propagation, and ultimately affecting the accuracy of height estimations (Frantz et al., 2021; Stilla et al., 2003)." (page 11-12, line 260-265)*

*Reference:*

*Frantz, D., Schug, F., Okujeni, A., Navacchi, C., Wagner, W., van der Linden, S., and Hostert, P.: National-scale*

*mapping of building height using Sentinel-1 and Sentinel-2 time series, Remote Sensing of Environment, 252, 112128-112128, https://doi.org/10.1016/j.rse.2020.112128, 2021.*

*Stilla, U., Soergel, U., and Thoennessen, U.: Potential and limits of InSAR data for building reconstruction in built-up areas, ISPRS Journal of Photogrammetry and Remote Sensing, 58, 113-123, https://doi.org/10.1016/S0924-2716(03)00021-2, 2003.*

---

## Author Comment (AC3)

**Reviewer #3 (Response to Reviewer)**

The study by Che et al. represents a significant advancement in the field of urban geography and Earth observations. The development of the 3D-GloBFP dataset is a groundbreaking achievement that fills a critical gap in the availability of global, high-resolution, and accurate building height information. The methodology employed is innovative and rigorous, resulting in a dataset with exceptional performance and reliability. The implications and applications of the 3D-GloBFP dataset are vast, spanning from climate modeling to sustainable development policies. Overall, this study deserves high praise for its contributions to the scientific community and beyond. However, to further strengthen the research, I would suggest addressing the following minor issues:

**Response**: thank you very much for your positive comments. We have carefully revised our manuscript based on your comments. We provided a detailed response to your comments below.

**Comment #1:** The division of the 33 regions mentioned in the paper is not particularly clear and requires a brief elaboration or a reference to the specific figure where they are illustrated.

**Response**: thank you for your suggestion. We included Figure 3, which shows the distribution of subregions and the number of training samples within each:

*"We divided the globe into 33 regions and developed the building height estimation model for each region, considering the non-uniform spatial distribution of samples and the heterogeneous building heights. Firstly, we divided the globe into 13 regions based on geographic spatial distance and regional development levels to ensure that each region has enough samples to train effective models. For instance, the Central and West Asian countries were considered as a single region for model training and estimation with 40040 training samples. However, given China's complex urban 3D structure and significant building heterogeneity (Wu et al., 2023), we further divided China into 21 regions. We built a separate height regression model for each region to ensure the effectiveness of the height estimation. For instance, considering the inadequacy of samples in Northwest China, we considered the provinces in Northwest as a single region with 8050 training samples for model training. Additionally, we considered the Beijing-Tianjin-Hebei, Yangtze River Delta, and Peral River Delta urban agglomerations as three separate regions due to the comparable economic levels and population size." (page 8-9, line 186-195)*

[Figure]

*Figure 3. Distribution of subregions.*

*Reference:*

*Wu, W.-B., Ma, J., Banzhaf, E., Meadows, M. E., Yu, Z.-W., Guo, F.-X., Sengupta, D., Cai, X.-X., and Zhao, B.: A first Chinese building height estimate at 10 m resolution (CNBH-10 m) using multi-source earth observations and machine learning, Remote Sensing of Environment, 291, 113578, https://doi.org/10.1016/j.rse.2023.113578, 2023.*

**Comment #2:** The first sentence of several paragraphs in the result section introduces the methodology, but it is recommended to revise them to summarize the findings of the current paragraph instead. You may place the corresponding methodology in the method section, or write it after the figure caption.

**Response**: thank you for your comments. We checked the first sentences of all the paragraphs in the result section and summarized the findings at the beginning of paragraphs, including:

*"Our 3D-GloBFP showed the most similar numerical distribution patterns to the reference heights across the United States, China, and Europe (Fig. 7)." (page 16, line 343-344)*

*"In the respective regional comparisons, first, we found that our 3D-GloBFP outperforms other building-scale height datasets in the US." (page 17, line 354-355)*

*"Second, our 3D-GloBFP is similar to the reference height in terms of distribution and spatial patterns in China." (page 18, line 379)*

The description of methodology was reorganized in method section.

**Comment #3:** While most of the results presented in this paper appropriately utilize the present simple tense, there are some instances where the past tense has been used inappropriately, see for example, line 149.

**Response**: thank you for the comment. We checked the tense throughout the paper. We used the

simple past tense to describe the completed process and present the research findings. And the simple present tense is used to describe the contents of figures and tables. The language has been polished by professional editing. The revised sentences in the article include:

*"The validation results with interpreted heights from Google Earth Street Views indicated the estimated results are consistent with the reference heights in the metropolitans of countries around the world, particularly for those landmark buildings." (page 12, line 273)*

*"Our estimated building heights provided more details of urban morphology and show more accurate results compared to the other four existing global datasets." (page 14, line 289)*

*"The building volume and area of representative cities varied significantly across different regions worldwide." (page 22, line 438)*

**Comment #4:** Expanding the discussion of challenges and future work would provide valuable insights into the dataset's limitations and potential for growth.

**Response**: thank you for your suggestion. First, improving the accuracy of building height estimations across various scenarios remains a key challenge. For instance, in densely built urban cores with more high-rise buildings, the complex structures and dense spatial patterns can lead to an underestimation of high-rise buildings. Furthermore, generating three-dimensional building footprints with temporal information is also a challenge. The dataset reflects the building structure for only a single year, whereas building heights may vary over time due to new construction and the demolition of existing buildings. In future work, we plan to use LiDAR datasets and filed survey data to improve the accuracy of building height estimation, especially for high-rise buildings. Also, we aim to develop multi-temporal building datasets to capture the dynamic changes in three-dimensional urban landscapes.

We have revised the manuscript to include a more detailed examination of the challenges and the future work:

*"The resolution of coarse-resolution remote sensing dataset (e.g., DSM with a 30 m resolution and nighttime light with a 463.83 m resolution) make it difficult to capture the heterogeneity features of super tall buildings, especially in densely built urban cores. Moreover, height and material of high-rise buildings, as well as the side-looking scene illumination Sentinel sensor, can cause complex multipath effects, complicating radar signal propagation, and ultimately affecting the accuracy of height estimations (Frantz et al., 2021; Stilla et al., 2003)." (page 11, line 260-264)*

*"Second, the current version of 3D-GloBFP shows relatively lower accuracy in areas with limited building height samples (i.e., suburb of South America). Integrating additional data (i.e., ground survey data and LiDAR datasets) to create more representative samples can enhance the accuracy of building height estimation. Additionally, the current version of 3D-GloBFP represents building height of a single year (i.e., 2020), as the model inputs (i.e., multi-source datasets) were collected around that time. This temporal limitation restricts the dataset's ability to reflect changes over time. We are also committed to producing 3D building datasets with temporal information to capture the dynamic changes of urban landscape." (page 23-24, line 462-468)*

**Comment #5:** Identifying gaps in current knowledge, discussing opportunities for integrating additional data sources, and outlining plans for updating and maintaining the dataset over time

would demonstrate the authors' commitment to ongoing improvement and research.

**Response**: thank you for your insightful comments. The limitations of the dataset include the accuracy in areas with limited samples and the coverage in some countries. First, areas with fewer samples, primarily undeveloped areas or cities with inadequate data show lower precision. Second, spatial coverage of three-dimensional building height data is incomplete in some regions due to the lack of detailed building boundary extraction, particularly in rural areas of certain countries. To address these issues, we will integrate additional data sources, including ground survey data, LiDAR, and other publicly available remote sensing data to enhance the accuracy of building height estimation models. Additionally, we will regularly update the dataset as more building boundary datasets become available.

We have summarized the gaps and plans for integrating datasets and updating the proposed dataset to enhance the manuscript:

*"**4.6 Limitations and future work***

*While this study provides valuable insights, several limitations must be acknowledged. First, the coverage is limited in certain regions, leading to tiled spatial gaps within some countries. These gaps are due to the limited coverage of Microsoft building footprints at the time of data creation. As more building footprint datasets become available, we will continue to update and enhance 3D-GloBFP using comprehensive open-source data. Second, the current version of 3D-GloBFP shows relatively lower accuracy in areas with limited building height samples (i.e., suburb of South America). Integrating additional data (i.e., ground survey data and LiDAR datasets) to create more representative samples can enhance the accuracy of building height estimation." (page 23, line 458-464)*

By addressing these points, the authors can enhance the readability and presentation of their work, thereby ensuring that the 3D-GloBFP dataset becomes an invaluable asset to the scientific community.